# TransNormerLLM: A Faster and Better Large Language Model with Improved TransNormer

## Abstract

We present TransNormerLLM, the first linear attention-based Large Language Model (LLM) that outperforms conventional softmax attention-based models in terms of both accuracy and efficiency. TransNormerLLM evolves from the previous linear attention architecture TransNormer (Qin et al., 2022a) by making advanced modifications that include positional embedding, linear attention acceleration, gating mechanism, tensor normalization, and inference acceleration and stabilization. Specifically, we use LRPE (Qin et al., 2023b) together with an exponential decay to avoid attention dilution issues while allowing the model to retain global interactions between tokens. Additionally, we propose Lightning Attention, a cutting-edge technique that accelerates linear attention by more than twice in runtime and reduces memory usage by a remarkable four times. To further enhance the performance of TransNormer, we leverage a gating mechanism to smooth training and a new tensor normalization scheme to accelerate the model, resulting in an impressive acceleration of over 20%. Furthermore, we develop a robust inference algorithm that ensures numerical stability and consistent inference speed, regardless of the sequence length, showcasing superior efficiency during both training and inference stages. We also implement an efficient model parallel schema for TransNormerLLM, enabling seamless deployment on large-scale clusters and facilitating expansion to even more extensive models, *i.e.,* LLMs with 175B parameters. We validate our model design through a series of ablations and train models with sizes of 385M, 1B, and 7B on our self-collected corpus. Benchmark results demonstrate that our models not only match the performance of state-of-the-art LLMs with Transformer but are also significantly faster.

## 1 Introduction

The field of Natural Language Processing (NLP) has been revolutionized by the advent of large-scale language models (LLMs) (Touvron et al., 2023a; Biderman et al., 2023; Brown et al., 2020). These models have demonstrated exceptional performance across a multitude of tasks, elevating abilities to comprehend, generate, and interact with human languages in computational frameworks. Previous language modeling development has predominantly centered around Transformer architectures, with seminal models such as vanilla Transformer (Vaswani et al., 2017), GPT series (Radford et al., 2018; 2019; Brown et al., 2020), BERT (Devlin et al., 2019), and BART (Lewis et al., 2019) standing as standard backbones in related fields. The success of Transformer architectures is premised on the softmax attention mechanism, which discerns dependencies among input tokens in a data-driven scheme and has global position awareness, offering the model an effective way to handle the long-range dynamism of natural language.

Nevertheless, conventional Transformers are not without their constraints. Primarily, their quadratic time complexity with respect to the sequence length limits their scalability and hampers efficiency in terms of computational resources and time during the training and inference stages. Numerous efficient sequence modeling methods have been proposed in an attempt to reduce the quadratic time complexity to linear (Katharopoulos et al., 2020; Choromanski et al., 2021; Qin et al., 2022b; Zheng et al., 2023; 2022). However, there are two reasons that prohibit them to be applied to LLMs: 1) their performance in language modeling is often unsatisfactory; 2) they do not demonstrate speed advantages in real-world scenarios.

In this paper, we introduce TransNormerLLM, the first linear attention-based LLM that surpasses conventional softmax attention in both accuracy and efficiency. The development of TransNormerLLM builds upon the foundations of the previous linear attention architecture, TransNormer (Qin et al., 2022a), while incorporating a series of advanced modifications to achieve superior performance. The key enhancements in TransNormerLLM include positional embedding, linear attention acceleration, gating mechanism, tensor normalization, and inference acceleration.

One notable improvement is the replacement of the TransNormer's DiagAttention with Linear Attention to enhance global interactions. To address the issue of dilution, we introduced LRPE (Qin et al., 2023b) with exponential decay (Press et al., 2022; Qin et al., 2023a; Peng et al., 2023a). Lightning Attention, a novel technique that significantly accelerates linear attention during training is introduced, resulting in a more than two-fold improvement, while also reducing memory usage by four times with IO awareness. Furthermore, we simplified GLU and Normalization, with the latter leading to a 20% speedup. A robust inference algorithm ensures the stability of numerical values and constant inference speed, regardless of the sequence length, thereby enhancing the efficiency of our model during both training and inference stages.

We validate the efficacy of TransNormerLLM on our self-collected pre-train corpus that is more than 6TB in size and contains over 2 trillion tokens. We expand the original TransNormer model ranging from 385M to 175B parameters and benchmark models with sizes of 385M, 1B, and 7B. The benchmark results demonstrate that our models achieve competitive performance with existing state-of-the-art Transformer-based LLMs with similar sizes while also having faster inference speeds. We will open-source our pre-trained models, enabling researchers and practitioners to build upon our work and explore efficient transformer structures in LLMs.

## 2 RELATED WORK

### 2.1 TRANSFORMER-BASED LLMS

In recent years, the field of Large Language Models (LLMs) has experienced significant advancements. Adhering to the scaling laws (Kaplan et al., 2020), various LLMs with over 100 billion parameters have been introduced, such as GPT-3 (Brown et al., 2020), Gopher (Rae et al., 2022), PaLM (Chowdhery et al., 2022), GLM (Du et al., 2022) and *etc.*. More specialized models like Galactica (Taylor et al., 2022) have also emerged for specific domains like science. A notable development is Chinchilla (Hoffmann et al., 2022), an LLM model with 70 billion parameters that redefines these scaling laws, focusing on the number of tokens rather than model weights. Furthermore, LLaMA (Touvron et al., 2023a) has also sparked interest due to its promising performance and open-source availability. The discourse around LLMs also encompasses the dynamics between open-source and closed-source models. Open-source models such as BLOOM (Workshop et al., 2023), OPT (Zhang et al., 2022), LLaMA (Touvron et al., 2023a), Pythia (Biderman et al., 2023) and Falcon (Penedo et al., 2023) are rising to compete against their closed-source counterparts, including GPT-3 (Brown et al., 2020) and Chinchilla (Hoffmann et al., 2022). To speed up training, Sparse Attention (Child et al., 2019; Beltagy et al., 2020) was introduced, but among large models, only GPT-3 adopted it (Brown et al., 2020; Scao et al., 2022).

### 2.2 NON-TRANSFORMER-BASED LLMS CANDIDATES

Despite the proliferation of Transformer-based large models in the research community, a portion of recent work has prioritized addressing its square time complexity. This focus has led to the exploration and development of a series of model architectures that diverge from the traditional Transformer structure. Among them, four significant contenders—linear transformers, state space model, long convolution, and linear recurrence—have shown promising results as substitutes for self-attention (SA) modules when modeling long sequences. These alternatives are favored for their superior asymptotic time complexity and competitive performances.

**Linear Transformer** Linear Transformer decomposes Softmax Attention into the form of the inner product of hidden representations, which allows it to use the "Right Product Trick," where the product of keys and values is computed to avoid the quadratic $n \times n$ matrix. Different methods utilize various hidden representations. For example, Katharopoulos et al. (2020) use 1+elu as an activation function, Qin et al. (2022b) use the cosine function to approximate the properties of softmax, and Ke et al.

(2021); Zheng et al. (2022; 2023) approximate softmax through theoretical approaches. Although its theoretical complexity is $O(nd^2)$, the actual computational efficiency of Linear Attention becomes quite low when used in causal attention due to the need for *cumsum* operations (Hua et al., 2022). On the other hand, most Linear Transformers still exhibit a certain performance gap compared to traditional Transformers (Katharopoulos et al., 2020; Liu et al., 2022).

**State Space Model**  State Space Model is based on the State Space Equation for sequence modeling (Gu et al., 2022b), using special initialization (Gu et al., 2020; 2022a), diagonalization assumptions (Gupta et al., 2022), and some techniques (Dao et al., 2022b) to achieve performance comparable to Transformers. On the other hand, due to the characteristics of the State Space Equation, it enables inference to be conducted within constant complexity (Gu et al., 2022b).

**Long Convolution**  Long convolution models (Qin et al., 2023a; Fu et al., 2023) utilize a kernel size equal to the input sequence length, facilitating a wider context compared to traditional convolutions. Training these models involves the efficient $O(n \log n)$ Fast Fourier Transforms (FFT) algorithm. However, long convolutions pose certain challenges, such as the need for causal convolution inference, which necessitates caching all historical computations similar to SA's key-value (KV) cache. The memory requirements for handling long sequences, coupled with the higher inference complexity compared to RNNs, make them less ideal for processing long sequences.

**Linear RNN**  Linear RNNs (Orvieto et al., 2023; Peng et al., 2023b), in contrast, stand out as more suitable replacements for SA in long-sequence modeling. A notable example is the RWKV (Peng et al., 2023b) model, a linear RNN-based LLM that has shown competitive performance against similarly scaled GPT models.

## 3  TRANSNORMERLLM

### 3.1  ARCHITECTURE IMPROVEMENT

In this section, we thoroughly investigate each module of the network and propose several improvements to achieve an optimal balance between efficiency and performance. Below, we outline the key designs of each block along with the inspiration behind each change. For the details of configurations for TransNormerLLM variants from 385M to 175B parameters, see Appendix A.

#### 3.1.1  IMPROVEMENT 1: POSITION ENCODING

In TransNormer, DiagAttention is used at the lower layers to avoid dilution issues. However, this leads to a lack of global interaction between tokens. In TransNormerLLM, we leverage LRPE (Qin et al., 2023b) with exponential decay (Press et al., 2022; Qin et al., 2023a; Peng et al., 2023b) to address this issue, retaining full attention at the lower layers. The expression of our position encoding is as follows:

$$a_{st} = \mathbf{q}_s^\top \mathbf{k}_t \lambda^{s-t} \exp^{i\theta(s-t)}. \tag{1}$$

which we call LRPE-d - Linearized Relative Positional Encoding with exponential decay. Similar to the original LRPE, we set $\theta$ to be learnable. We empirically find that rather than applying LRPE-d to every layer, applying it to the first layer and keeping other layers with exponential decay can speed up training by approximately 15-20% but only with a subtle effect on the performance.

Note that this position encoding is fully compatible with Linear Attention, as it can be decomposed with respect to $s$ and $t$ separately. The value of $\lambda$ for the $h$-th head in the $l$-th layer (assuming there are a total of $H$ heads and $L$ layers) is given by:

$$\lambda = \exp\left(-\frac{8h}{H} \times \left(1 - \frac{l}{L}\right)\right). \tag{2}$$

Here, $\frac{8h}{H}$ corresponds to the decay rate of the $h$-th head, while $\left(1 - \frac{l}{L}\right)$ corresponds to the decay rate of the $l$-th layer. The term $\left(1 - \frac{l}{L}\right)$ ensures that the Theoretical Receptive Fields (TRF) (Qin et al., 2023c) at the lower layers is smaller compared to the higher layers, which aligns with TransNormer's motivation. It should be noted that the decay rate in the last layer is set to 1, allowing each token to attend to global information. We choose $\lambda$ to be non-learnable since we empirically found that gradients become unstable when $\lambda$ is learnable, leading to NaN values.

### 3.1.2 IMPROVEMENT 2: GATING MECHANISM

Gate can enhance the performance of the model and smooth the training process. In TransNormer-LLM, we adopted the approach from Flash (Hua et al., 2022) and used the structure of Gated Linear Attention (GLA) in token mixing:

$$\text{TokenMixer} : \mathbf{O} = \text{Norm}(\mathbf{Q}\mathbf{K}^\top \mathbf{V}) \odot \mathbf{U}, \tag{3}$$

where:
$$\mathbf{Q} = \phi(\mathbf{X}\mathbf{W}_q), \mathbf{K} = \phi(\mathbf{X}\mathbf{W}_k), \mathbf{V} = \mathbf{X}\mathbf{W}_v, \mathbf{U} = \mathbf{X}\mathbf{W}_u. \tag{4}$$

We choose $\phi$ to be swish (Ramachandran et al., 2017) activation function as we empirically find that it outperforms other activation functions, as shown in Table 6.

To further accelerate the model, we propose Simple GLU (SGLU), which removes the activation function from the original GLU structure as the gate itself can introduce non-linearity. Therefore, our channel mixing becomes:

$$\text{ChannelMixer} : \mathbf{O} = [\mathbf{V} \odot \mathbf{U}]\mathbf{W}_o, \mathbf{V} = \mathbf{X}\mathbf{W}_v, \mathbf{U} = \mathbf{X}\mathbf{W}_u, \tag{5}$$

We empirically find that not using an activation function in GLU will not lead to any performance loss, as demonstrated in Table 7.

### 3.1.3 IMPROVEMENT 3: TENSOR NORMALIZATION

We employ the NormAttention introduced in TransNormer (Qin et al., 2022a) as follows:

$$\mathbf{O} = \text{Norm}((\mathbf{Q}\mathbf{K}^\top)\mathbf{V}) \tag{6}$$

This attention mechanism eliminates the softmax and scaling operation. Moreover, it can be transformed into linear attention through right multiplication:

$$\mathbf{O} = \text{Norm}(\mathbf{Q}(\mathbf{K}^\top \mathbf{V})) \tag{7}$$

This linear form allows for recurrent prediction with a complexity of $O(nd^2)$, making it efficient during inference. Specifically, we only update $\mathbf{K}^\top \mathbf{V}$ in a recurrent manner without computing the full attention matrix. In TransNormerLLM, we replace the RMSNorm with a new simple normalization function called SimpleRMSNorm, abbreviated as SRMSNorm:

$$\text{SRMSNorm}(\mathbf{x}) = \frac{\mathbf{x}}{\|\mathbf{x}\|_2/\sqrt{d}}. \tag{8}$$

We empirically find that using SRMSNorm does not lead to any performance loss, as demonstrated in the ablation study in Table. 8.

### 3.1.4 THE OVERALL STRUCTURE

The overall structure is illustrated in Figure 1. In this structure, the input $\mathbf{X}$ is updated through two consecutive steps: First, it undergoes Gated Linear Attention (GLA) with the application of SimpleRMSNorm (SRMSNorm) normalization. Then, it goes through the Simple Gated Linear Unit (SGLU) with SRMSNorm normalization again. This overall architecture helps improve the model's performance based on the PreNorm approach. The pseudo-code of the overall process is as follows:

$$\begin{aligned} \mathbf{X} &= \mathbf{X} + \text{GLA}(\text{SRMSNorm}(\mathbf{X})), \\ \mathbf{X} &= \mathbf{X} + \text{SGLU}(\text{SRMSNorm}(\mathbf{X})). \end{aligned} \tag{9}$$

## 3.2 TRAINING OPTIMIZATION

### 3.2.1 LIGHTNING ATTENTION

The structure of linear attention allows for efficient attention calculation with a complexity

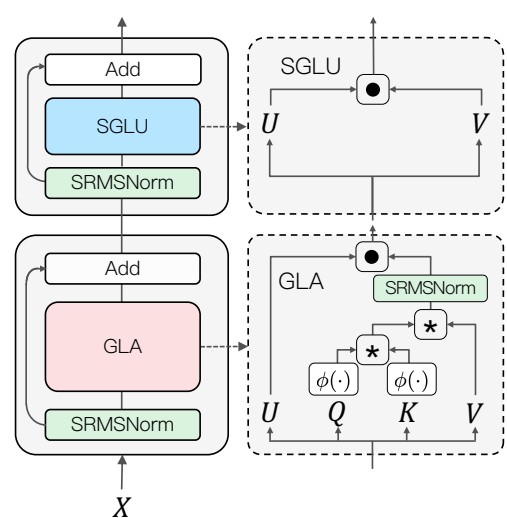

Figure 1: Architecture overview of the proposed model. Each transformer block is composed of a Gated Linear Attention(GLA) for token mixing and a Simple Gated Linear Unit (SGLU) for channel mixing. We apply pre-norm for both modules.

of $O(nd^2)$ through right-multiplication. However, for causal prediction, right-multiplication is not efficient as it necessitates *cumsum* computation (Hua et al., 2022), which hinders parallelism training. As a result, during training, we continue to use the conventional left-multiplication version. To accelerate attention calculations, we introduce the Lightning Attention algorithm inspired by (Dao, 2023; Dao et al., 2022a), which makes our linear attention IO-friendly. It computes the following:

$$\mathbf{O} = (\mathbf{Q}\mathbf{K}^\top \odot \mathbf{M})\mathbf{V}. \tag{10}$$

Here, $\mathbf{M}$ is the attention mask which enables lower triangular causal masking and positional encoding. In the Lightning Attention, we split the inputs $\mathbf{Q}, \mathbf{K}, \mathbf{V}$ into blocks, load them from slow HBM to fast SRAM, then compute the attention output with respect to those blocks. Then we accumulate the final results. The computation speed is accelerated by avoiding the operations on slow HBM. The implementation details of Lightning Attention are shown in Appendix B, where Algorithm 3 for forward pass and Algorithm 4 for backward pass.

### 3.2.2 MODEL PARALLELISM ON TRANSNORMERLLM

To effectively execute large-scale pre-training for TransNormerLLM, we have put efforts on system optimization encompassing various dimensions. Specifically, we employ fully sharded data parallelism (FSDP) (Zhao et al., 2023), a technique that shards all model parameters, gradients, and optimizer state tensors across the entire cluster. This strategic partition significantly reduces the memory footprint on each individual GPU, thereby enhancing memory utilization. In our pursuit of greater efficiency, we leverage activation checkpointing (Shoeybi et al., 2019), which minimizes the cached activations in memory during the forward pass. Instead of retaining these activations, they are recomputed when calculating gradients in the backward pass. This approach saves huge GPU memory thus enable to apply bigger batch size. Furthermore, we harness automatic mixed precision (AMP) (Micikevicius et al., 2017) to simultaneously save GPU memory and expedite computational speed. It's noteworthy that in our experimental setup, we employ BFloat16 (Kalamkar et al., 2019) due to its observed advantage in enhancing the training stability of TransNormerLLM models.

In addition to the previously mentioned optimization endeavors, we delve deeper into the realm of system engineering by implementing model parallelism specifically tailored to linear transformers, drawing inspiration from Megatron-LM model parallelism (Shoeybi et al., 2019). In a standard transformer model, each transformer layer comprises a self-attention block followed by a two-layer multi-layer perceptron (MLP) block. Megatron-LM model parallelism independently addresses these two constituent blocks. Similarly, within the architecture of TransNormerLLM, characterized by its two primary components, SGLU and GLA, we apply model parallelism to each of these components separately. The intricate details of our model parallelism strategies are elaborated below.

**Model Parallelism on SGLU**   Recall the SGLU structure in (5):

$$\mathbf{O} = [(\mathbf{X}\mathbf{W}_v) \odot (\mathbf{X}\mathbf{W}_u)]\mathbf{W}_o, \tag{11}$$

The model parallelism adaptation of SGLU is as follows:

$$[\mathbf{O}_1', \mathbf{O}_2'] = \mathbf{X}[\mathbf{W}_v^1, \mathbf{W}_v^2] \odot \mathbf{X}[\mathbf{W}_u^1, \mathbf{W}_u^2], = [\mathbf{X}\mathbf{W}_v^1, \mathbf{X}\mathbf{W}_v^2] \odot [\mathbf{X}\mathbf{W}_u^1, \mathbf{X}\mathbf{W}_u^2], \tag{12}$$

which splits the weight matrices $\mathbf{W}_v$ and $\mathbf{W}_u$ along their columns and obtains an output matrix splitting along its columns too. Then the split output $[\mathbf{O}_1, \mathbf{O}_2]$ is multiplied by another matrix which is split along its rows as:

$$\mathbf{O} = [\mathbf{O}_1', \mathbf{O}_2'][\mathbf{W}_o^1, \mathbf{W}_o^2]^\top = \mathbf{O}_1'\mathbf{W}_o^1 + \mathbf{O}_2'\mathbf{W}_o^2 \tag{13}$$

Similar with model parallelism in Megatron-LM, this whole procedure splits three general matrix multiplies (GEMMs) inside the SGLU block across multiple GPUs and only introduces a single *all-reduce* collective communication operation in both the forward and backward passes, respectively.

**Model Parallelism on GLA**   Recall the GLA block in (3) and (4), its model parallelism version is:

$$[\mathbf{O_1}, \mathbf{O_2}] = \text{SRMSNorm}(\mathbf{Q}\mathbf{K}^\top\mathbf{V}) \odot \mathbf{U}, \tag{14}$$

where:

$$\mathbf{Q} = [\phi(\mathbf{X}\mathbf{W}_q^1), \phi(\mathbf{X}\mathbf{W}_q^2)], \mathbf{K} = [\phi(\mathbf{X}\mathbf{W}_q^1), \phi(\mathbf{X}\mathbf{W}_q^2)], \mathbf{V} = \mathbf{X}[\mathbf{W}_v^1, \mathbf{W}_v^2], \mathbf{U} = \mathbf{X}[\mathbf{W}_u^1, \mathbf{W}_u^2], \tag{15}$$

Note that in our implementation, we use the combined QKVU projection to improve computation efficiency for linear attention. The obtained split output matrix $[\mathbf{O_1}, \mathbf{O_2}]$ again is multiplied by a weight matrix split along its columns which is similar to (13).

**Algorithm 1** Origin Inference Algorithm

> **Input:** $\mathbf{q}_t, \mathbf{k}_t, \mathbf{v}_t, t = 1, \ldots, n;$
> **Output:** $\mathbf{o}_t, t = 1, \ldots, n;$
> **Initialize:** $[\mathbf{kv}]_0 = \mathbf{0};$
> **for** $t = 1, \ldots, n$ **do**
>    $[\mathbf{kv}]_t = [\mathbf{kv}]_{t-1} + \mathbf{k_t} \lambda^{-t} \mathbf{v}_t^\top,$
>    $\mathbf{o}_t = \mathbf{q}_t \lambda^t [\mathbf{kv}]_t.$
> **end for**

**Algorithm 2** Robust Inference Algorithm

> **Input:** $\mathbf{q}_t, \mathbf{k}_t, \mathbf{v}_t, t = 1, \ldots, n;$
> **Output:** $\mathbf{o}_t, t = 1, \ldots, n;$
> **Initialize:** $[\overline{\mathbf{kv}}]_0 = \mathbf{0};$
> **for** $t = 1, \ldots, n$ **do**
>    $[\overline{\mathbf{kv}}]_t = \lambda [\overline{\mathbf{kv}}]_{t-1} + \mathbf{k_t} \mathbf{v}_t^\top,$
>    $\mathbf{o}_t = \mathbf{q}_t [\overline{\mathbf{kv}}]_t.$
> **end for**

## 3.3 ROBUST INFERENCE

In this section, we discuss the inference problem in TransNormerLLM. It is important to note that the formula 1 can be decomposed into the following form:

$$a_{st} = (\mathbf{q}_s \lambda^s \exp^{i\theta s})^\top (\mathbf{k}_t \lambda^{-t} \exp^{i\theta t}). \tag{16}$$

This allows TransNormerLLM to perform inference in the form of an RNN. Details of the procedure are shown in Algorithm 1. However, it is worth noting that $\lambda < 1$, which results in:

$$\|\mathbf{q}_s \lambda^s \exp^{i\theta s}\|_2 = \|\mathbf{q}_s\|_2 \lambda^s \to 0, \|\mathbf{k}_t \lambda^{-t} \exp^{i\theta t}\|_2 = \|\mathbf{k}_t\|_2 \lambda^{-t} \to \infty, \tag{17}$$

leading to numerical precision issues.

To avoid these issues, we propose a Robust Inference Algorithm in 2. Since $\|\mathbf{q}_s \exp^{i\theta s}\| = \|\mathbf{q}_s\|$, $\|\mathbf{k}_t \exp^{i\theta t}\| = \|\mathbf{k}_t\|$, for simplicity, we will omit LRPE (Qin et al., 2023b) in the subsequent discussions, considering only $a_{st} = \mathbf{q}_s^\top \mathbf{k}_t \lambda^{s-t}$. We provide a mathematical proof of $[\mathbf{kv}]_t = \lambda^{-t}[\overline{\mathbf{kv}}]_t$ in Appendix C

## 4 EXPERIMENTS

We use PyTorch (Paszke et al., 2019) and Triton (Tillet et al., 2019) to implement TransNormerLLM in Metaseq framework (Zhang et al., 2022). Our model is trained using Adam optimizer (Kingma & Ba, 2017), and we employ FSDP to efficiently scale our model to NVIDIA A100 80G clusters. We additionally leverage the model parallel as appropriate to optimize performance. In ablation studies, all models are trained on a sampled corpus from our corpus with 300B tokens. In order to reduce the fluctuation of Losses and PPLs in the tables below, we compute the average Losses and PPLs of the last 1k iterations as the final metrics. For our benchmark models, we train our 385M, 1B, and 7B models on our corpus for 1 trillion, 1.2 trillion, and 1.4 trillion tokens respectively. We use an input sequence length of 8192 tokens in our pretraining process. For a comprehensive understanding of our corpus, encompassing intricate details such as data preprocessing methods and tokenization procedures, we direct interested readers to Appendix D.

### 4.1 ARCHITECTURE ABLATIONS

**Transformer *vs* TransNormerLLM** We carried out a meticulous series of comparative tests between our TransNormerLLM and Transformer, spanning over an array of disparate sizes. The comparative performance of these models is clearly illustrated in Table 1. Under identical configurations, it becomes evident that our TransNormerLLM exhibits a superior performance profile compared to Transformer. We observed that TransNormerLLM outperformed Transformer by a remarkable 5% at the size of 385M. More importantly, as the size reached 1B, this superiority became even more pronounced, with an advantage of 9% for TransNormerLLM over Transformer.

Table 1: **Transformer *vs* TransNormerLLM.** TransNormerLLM performs better than Transformer in size of 385M and 1B under identical configurations by 5% and 9%, respectively.

| Model Size | 385M | | | 1B | | |
|---|---|---|---|---|---|---|
| Method | Updates | Loss | PPL | Updates | Loss | PPL |
| Transformer | 100K | 2.362 | 5.160 | 100K | 2.061 | 4.765 |
| TransNormerLLM | 100K | 2.248 | 4.770 | 100K | 1.896 | 3.729 |

**TransNormer *vs* TransNormerLLM** We compare the original TransNormer and the improved TransNormerLLM and the results are shown in Table 2. TransNormerLLM exhibited an enhancement of 2% and 1% respectively.

Table 2: **TransNormer *vs* TransNormerLLM.**

| Method | Params | Updates | Loss | PPL |
|---|---|---|---|---|
| TransNormerLLM | 385M | 100K | 2.248 | 4.770 |
| TransNormer-T1 | 379M | 100K | 2.290 | 4.910 |
| TransNormer-T2 | 379M | 100K | 2.274 | 4.858 |

**Positional Encoding** In the positional encoding experiment, we conducted a series of tests, comparing Mix (LRPE-d for the first layer, Exp-Decay for the rest), APE (Absolute Positional Encoding), LRPE, Exp-Decay (Exponential Decay), and LRPE-d. As evident from Table 3, Ours and LRPE-d achieve better performance than other options. We select the Mix positional encoding as it boosts the training speed up to 20% while only slightly worse than LRPE-d.

We also perform ablations on the decay temperature $\left(1 - \frac{l}{L}\right)$ in Eq. 2. The perplexity of the TransNormerLLM is reduced by adding the decay temperature, as shown in Table 4.

**Gating Mechanism** We conduct ablation studies to examine the effect of including the gating mechanism. As observed in Table 5, gate enabled the reduction of the loss value from 2.263 to 2.248.

**GLA Activation Functions** We conducted experiments on the GLA (Gated Linear Attention) structure with respect to the activation function. As shown in Table 6, using Swish and 1+elu leads to similar performance. However, in our experiments, using 1+elu in our 7B model may encounter a NaN problem, so we use Swish in our model.

**GLU Activation Functions** We conduct an experiment by removing the activation function within the Gated Linear Units (GLU) structure. As shown in Table 7, the results reveal that this alteration had a negligible impact on the final outcome. As a result, we decide to adopt the Simple Gated Linear Units (SGLU) structure in our final model configuration.

**Normalization functions** In our study, we conducted a series of ablation tests employing various normalization methods including SRM-SNorm, RMSNorm and LayerNorm. The results indicate that there is almost no difference among these methods when applied to TransNormer-LLM. Nevertheless, during the course of our testing, we revisited and re-engineered the SRM-SNorm using Triton. As it is shown in Figure 2, empirical evidence supports that our modification offers a significant boost in computational speed when operating with larger dimensions, compared to the PyTorch implementation methods.

Table 3: **Positional encoding.** LRPE-d leads to the most optimal outcome.

| PE Methods | Params | Updates | Loss | PPL |
|---|---|---|---|---|
| Mix | 385M | 100K | 2.248 | 4.770 |
| APE | 386M | 100K | 2.387 | 5.253 |
| Exp-Decay | 385M | 100K | 2.267 | 4.834 |
| LRPE | 385M | 100K | 2.287 | 4.899 |
| LRPE-d | 385M | 100K | 2.236 | 4.728 |

Table 4: **Ablations on decay temperature.** The results of decay temperature proved to be superior.

| Temperature | Params | Updates | Loss | PPL |
|---|---|---|---|---|
| w/ temperature | 385M | 100K | 2.248 | 4.770 |
| w/o temperature | 385M | 100K | 2.258 | 4.804 |

Table 5: **Ablations on gating mechanism.** The performance with the gate proved to be superior.

| Gate | Params | Updates | Loss | PPL |
|---|---|---|---|---|
| w/ gate | 385M | 100K | 2.248 | 4.770 |
| w/o gate | 379M | 100K | 2.263 | 4.820 |

Table 6: **Ablations on GLA activation functions.** The results obtained from different activation functions were virtually identical.

| GLA Act | Params | Updates | Loss | PPL |
|---|---|---|---|---|
| Swish | 385M | 100K | 2.248 | 4.770 |
| No Act | 385M | 100K | 2.283 | 4.882 |
| 1+elu | 385M | 100K | 2.252 | 4.767 |

Table 7: **Ablations on GLU activation functions.** The exclusion of the activation function had no negative impact on the results.

| GLU Act | Params | Updates | Loss | PPL |
|---|---|---|---|---|
| No Act | 385M | 100K | 2.248 | 4.770 |
| Swish | 385M | 100K | 2.254 | 4.788 |

Table 8: **Normalization Functions.** The deviation in results among the bellowing normalization functions is minimal.

| Norm Type | Params | Updates | Loss | PPL |
|---|---|---|---|---|
| SRMSNorm | 385M | 100K | 2.248 | 4.770 |
| RMSNorm | 385M | 100K | 2.247 | 4.766 |
| LayerNorm | 385M | 100K | 2.247 | 4.765 |

**Lightning Attention** We conducted a speed and memory comparison between our Lightning Attention and the baseline, which is the PyTorch implementation of the NormAttention (Qin et al., 2022a). Figure 3 (left) reports the runtime in milliseconds of the forward + backward pass. Baseline runtime grows quadratically with sequence length, while Lightning Attention operates significantly faster, at least $2\times$ faster than the PyTorch implementation. Figure 3 (right) reports the memory footprint of Lightning Attention compared to the baseline. The memory footprint of Lightning Attention grows linearly with sequence length, which is up to $4\times$ more efficient than the baseline when the sequence length is 8192. Our proposed Lightning Attention achieves superior efficiency.

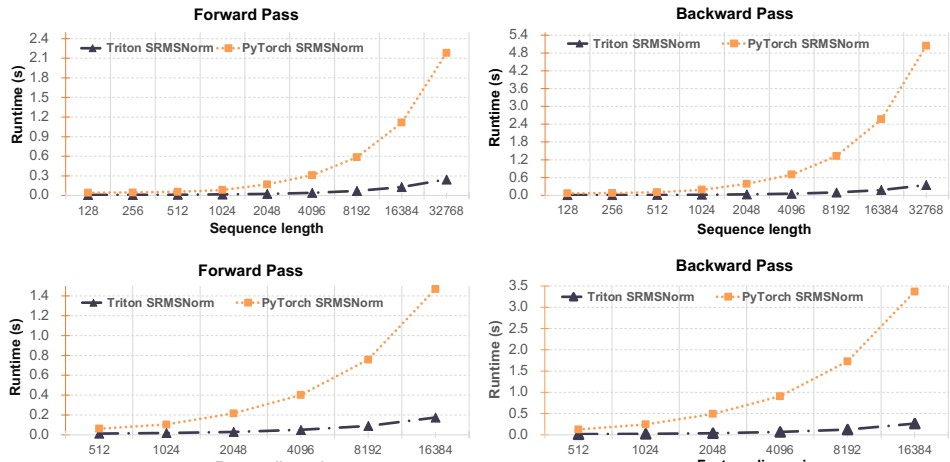

Figure 2: **Performance Evaluation of SRMSNorm Implementation.** The upper figures exhibit the runtime comparison of the forward pass (left) and backward pass (right) for different sequence lengths, with a fixed feature dimension of 3072. The lower two figures illustrate the runtime comparison for various feature dimensions, with a fixed sequence length of 4096.

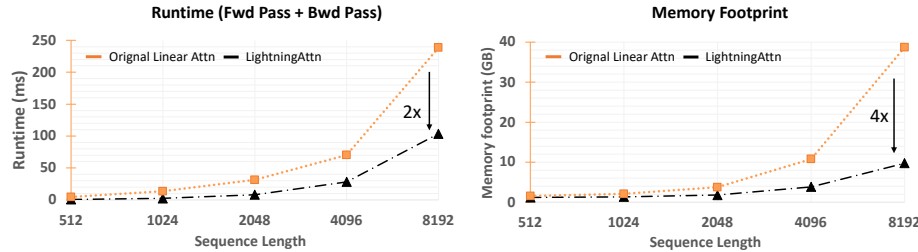

Figure 3: **Memory and speed comparison between linear attention and lightning attention.** Left: runtime of forward + backward pass milliseconds for different sequence lengths, with a fixed feature dimension of 2048. Right: memory footprints of forward + backward pass for different sequence lengths, with a fixed feature dimension of 2048.

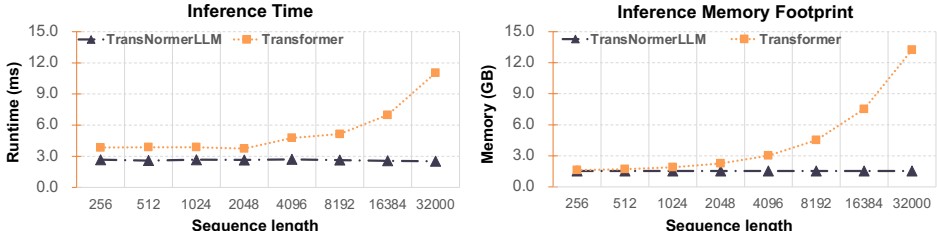

Figure 4: **Inference Time and Memory Footprint.** Left: inference runtime measured in milliseconds across different sequence lengths. Right: memory consumption during inference for varying sequence lengths. It is noteworthy that as the sequence length increases, TransNormerLLM demonstrates a consistent inference time and memory footprint.

## 4.2 BENCHMARKS

In order to validate the effectiveness of TransNormerLLM, we tested our 385M, 1B, and 7B models on Commonsense Reasoning Task, MMLU(Hendrycks et al., 2021), CMMLU(Li et al., 2023), and C-Eval(Huang et al., 2023). For comparison, we selected several open-source models as competitors, including Transformer-based models such as OPT (Zhang et al., 2022), Pythia (Biderman et al., 2023), BLOOM (Workshop et al., 2023), GPT-Neo (Black et al., 2022), GPT-J (Wang & Komatsuzaki, 2021), MPT (Team et al., 2023), Falcon (Almazrouei et al., 2023), LLaMA1/2 (Touvron et al., 2023a;b), OpenLLAMA v1/v2 (Geng & Liu, 2023), Baichuan 1/2 (Baichuan, 2023), ChatGLM 1/2 (Zeng et al., 2022; Du et al., 2022), and non-Transformer model RWKV (Peng et al., 2023a). It can be observed that, compared to these models, TransNormerLLM remains highly competitive.

Table 9: **Performance Comparison on Commonsense Reasoning and Aggregated Benchmarks.**
For a fair comparison, we report competing methods' results reproduced by us using their released models. Official results are denoted in *italics*. PS: parameter size (billion). T: tokens (trillion). HS: HellaSwag. WG: WinoGrande.

| Model | PS | T | BoolQ | PIQA | HS | WG | ARC-e | ARC-c | OBQA | MMLU | CMMLU | C-Eval |
|---|---|---|---|---|---|---|---|---|---|---|---|---|
| OPT | 0.35 | 0.30 | 57.74 | 64.58 | 36.69 | 52.49 | 44.02 | 23.89 | 28.20 | 26.02 | 25.34 | 25.71 |
| Pythia | 0.40 | 0.30 | 60.40 | 67.08 | 40.52 | 53.59 | 51.81 | 24.15 | 29.40 | 25.99 | 25.16 | 24.81 |
| BLOOM | 0.56 | 0.35 | 55.14 | 64.09 | 36.97 | 52.80 | 47.35 | 23.98 | 28.20 | 24.80 | 25.35 | 27.14 |
| RWKV | 0.43 | 0.30 | - | *67.52* | *40.90* | *51.14* | *52.86* | *25.17* | *32.40* | 24.85 | - | - |
| Ours | 0.39 | 1.0 | 62.14 | 66.70 | 46.27 | 54.46 | 55.43 | 27.99 | 32.40 | 25.90 | 25.05 | 25.24 |
| GPT-Neo | 1.3 | 0.3 | 61.99 | 71.11 | 48.93 | 54.93 | 56.19 | 25.85 | 33.60 | 24.82 | 26.03 | 23.94 |
| OPT | 1.3 | 0.3 | 57.77 | 71.71 | 53.70 | 59.35 | 57.24 | 29.69 | 33.20 | 24.96 | 24.97 | 25.32 |
| Pythia | 1.4 | 0.3 | 60.73 | 70.67 | 47.18 | 53.51 | 56.99 | 26.88 | 31.40 | 26.55 | 25.13 | 24.25 |
| BLOOM | 1.1 | 0.35 | 59.08 | 67.14 | 42.98 | 54.93 | 51.47 | 25.68 | 29.40 | 27.30 | 25.09 | 26.50 |
| RWKV | 1.5 | 0.3 | - | *72.36* | *52.48* | *54.62* | *60.48* | *29.44* | *34.00* | 25.77 | - | - |
| Falcon | 1.0 | 0.35 | 61.38 | 75.14 | 61.50 | 60.30 | 63.38 | 32.17 | 35.60 | 25.28 | 24.88 | 25.66 |
| Ours | 1.0 | 1.2 | 63.27 | 72.09 | 56.49 | 60.38 | 63.68 | 35.24 | 36.60 | 27.10 | 25.88 | 26.01 |
| GPT-J | 6.9 | 0.3 | 65.44 | 75.41 | 66.25 | 64.09 | 66.92 | 36.60 | 38.20 | 25.40 | 26.47 | 23.39 |
| OPT | 6.7 | 0.3 | 66.18 | 76.22 | 67.21 | 65.19 | 65.66 | 34.64 | 37.20 | 24.57 | 25.36 | 25.32 |
| Pythia | 6.9 | 0.3 | 63.46 | 75.14 | 63.92 | 60.77 | 67.34 | 35.41 | 37.00 | 24.64 | 25.56 | 26.40 |
| BLOOM | 7.1 | 0.35 | 62.91 | 72.69 | 62.33 | 64.01 | 65.11 | 33.45 | 35.80 | 26.25 | 24.97 | 24.25 |
| RWKV | 7.4 | 0.3 | - | *76.06* | *65.51* | *61.01* | *67.80* | *37.46* | *40.20* | 24.96 | - | - |
| MPT | 6.9 | 1.0 | 73.88 | 79.43 | 76.25 | 68.27 | 74.79 | 41.72 | 42.20 | 30.80 | 25.99 | 24.06 |
| Falcon | 7.2 | 1.5 | 73.73 | 79.38 | 76.3 | 67.17 | 74.62 | 43.60 | 43.80 | 27.79 | 25.73 | 22.92 |
| Baichuan1 | 7.0 | 1.2 | 70.09 | 76.01 | 70.06 | 64.09 | 71.72 | 40.53 | 38.20 | *42.30* | *44.43* | *42.80* |
| Baichuan2 | 7.0 | 2.6 | 72.72 | 76.50 | 72.17 | 68.35 | 75.17 | 42.32 | 39.60 | *54.16* | *57.07* | *54.00* |
| ChatGLM1 | 6.7 | 1.0 | 74.74 | 68.88 | 45.57 | 52.25 | 48.78 | 31.66 | 36.80 | *40.63* | 37.48 | *40.23* |
| ChatGLM2 | 7.1 | 1.4 | 77.65 | 69.37 | 50.51 | 57.62 | 59.13 | 34.30 | 37.00 | *45.46* | 48.80 | *52.55* |
| OpenLLaMAv1 | 6.7 | 1.0 | 70.43 | 75.68 | 69.23 | 66.69 | 71.17 | 38.57 | 39.00 | 30.49 | 25.40 | 26.09 |
| OpenLLaMAv2 | 6.7 | 1.0 | 72.20 | 78.84 | 74.51 | 65.67 | 72.39 | 41.30 | 41.00 | 41.29 | 29.58 | 30.01 |
| LLaMA1 | 6.7 | 1.0 | *76.50* | *79.80* | *76.10* | *70.10* | *72.80* | *47.60* | *57.20* | *35.10* | 25.62 | 25.72 |
| LLaMA2 | 6.7 | 2.0 | *77.68* | *78.07* | *76.02* | *68.98* | *76.30* | *46.33* | *44.20* | *45.30* | 32.96 | 33.20 |
| Ours | 6.8 | 1.4 | 75.87 | 80.09 | 75.21 | 66.06 | 75.42 | 44.40 | 63.40 | 43.10 | 47.99 | 43.18 |

**Commonsense Reasoning** We report BoolQ (Clark et al., 2019), PIQA (Bisk et al., 2019), SIQA (Sap et al., 2019), HellaSwag (Zellers et al., 2019), WinoGrande (Sakaguchi et al., 2019), ARC easy and challenge (Clark et al., 2018), OpenBookQA (Mihaylov et al., 2018) and their average. We report 0-shot results for all benchmarks using LM-Eval-Harness (Gao et al., 2021). All of our models achieve competitive performance compared to existing state-of-the-art LLMs, showcasing a remarkable ability to comprehend and apply commonsense reasoning.

**Aggregated Benchmarks** We report the overall results for MMLU (Hendrycks et al., 2021), CMMLU (Li et al., 2023), C-Eval (Huang et al., 2023). Official scripts were used for evaluating MMLU, CMMLU, and C-Eval, with all evaluation results being conducted with a 5-shot setup. In comparison to top-tier open-source models available in the industry, our models have demonstrated matched performance in both English and Chinese benchmarks.

### 4.3 SCALING TO 175B

Furthermore, we have carried out a series of experiments to assess the efficacy of model parallelism as applied to the TransNormerLLM architecture. The comprehensive outcomes of these experiments have been thoughtfully presented in Appendix E.1. Moreover, our research extends to the meticulous evaluation of various cutting-edge system optimization techniques. This evaluation encompasses their impact on both training speed and context length across models ranging from 7B to 175B in scale. We have thoughtfully documented the detailed results of these experiments in Appendix E.2.

## 5 CONCLUSION

We introduced TransNormerLLM in this paper, an improved TransNormer that is tailored for LLMs. Our TransNormerLLM consistently outperformed Transformers in both accuracy and efficiency. Extensive ablations demonstrate the effectiveness of our modifications and innovations in position encoding, gating mechanism, activation functions, normalization functions, and lightning attentions. These modifications collectively contribute to TransNormerLLM's outstanding performance, positioning it as a promising choice for state-of-the-art language models. The benchmark results for models with sizes of 385 million, 1 billion, and 7 billion parameters unequivocally demonstrate that TransNormerLLM not only matches the performance of current leading Transformer-based Large Language Models (LLMs) but also enjoys faster inference speeds. We will release our pre-trained TransNormerLLM models to foster community advancements in efficient LLM.

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

# Appendix

## A  MODEL

We present distinct model variants of the TransNormerLLM architecture, delineating their respective configurations with regard to parameters, layers, attention heads, and hidden dimensions. The detailed specifications are meticulously tabulated in Table 10.

Table 10: **TransNormerLLM Model Variants.**

| Model Size | Non-Embedding Params | Layers | Hidden Dim | Heads | Equivalent Models |
|---|---|---|---|---|---|
| 385M | 384,974,848 | 24 | 1024 | 8 | Pythia-410M |
| 1B | 992,165,888 | 16 | 2048 | 16 | Pythia-1B |
| 3B | 2,876,006,400 | 32 | 2560 | 20 | Pythia-2.8B |
| 7B | 6,780,547,072 | 30 | 4096 | 32 | LLAMA-6.7B |
| 13B | 12,620,195,840 | 36 | 5120 | 40 | LLAMA-13B |
| 65B | 63,528,009,728 | 72 | 8192 | 64 | LLAMA-65B |
| 175B | 173,356,498,944 | 88 | 12288 | 96 | GPT-3 |

## B  LIGHTNING ATTENTION

We present the algorithm details of Lightning Attention includes forward pass and backward pass in Algorithm 3 and 4, respectively.

---

**Algorithm 3** Lightning Attention Forward Pass

---

**Input:** $\mathbf{Q}, \mathbf{K}, \mathbf{V} \in \mathbb{R}^{n \times d}$, attention mask $\mathbf{M} \in \mathbb{R}^{n \times n}$, block sizes $B_c, B_r$;
**Initialize:** $\mathbf{O} = \mathbf{0} \in \mathbb{R}^{n \times d}$;
Divide $\mathbf{Q}$ into $T_r = \frac{n}{B_r}$ blocks $\mathbf{Q}_1, \mathbf{Q}_2, ... \mathbf{Q}_{T_r}$ of size $B_r \times d$ each.
Divide $\mathbf{K}, \mathbf{V}$ into $T_c = \frac{n}{B_c}$ blocks $\mathbf{K}_1, \mathbf{K}_2, ... \mathbf{K}_{T_c}, \mathbf{V}_1, \mathbf{V}_2, ... \mathbf{V}_{T_c}$ of size $B_c \times d$ each.
Divide $\mathbf{O}$ into $T_r = \frac{n}{B_r}$ blocks $\mathbf{O}_1, \mathbf{O}_2, ... \mathbf{O}_{T_r}$ of size $B_r \times d$ each.
Divide $\mathbf{M}$ into $T_r \times T_c$ blocks $\mathbf{M}_{11}, \mathbf{M}_{12}, ... \mathbf{M}_{T_r,T_c}$ of size $B_r \times B_c$ each.
**for** $1 \le i \le T_r$ **do**
    Load $\mathbf{Q}_i \in \mathbb{R}^{B_r \times d}$ from HBM to on-chip SRAM.
    Initialize $\mathbf{O}_i = \mathbf{0} \in \mathbb{R}^{B_r \times d}$ on SRAM.
    **for** $1 \le j \le T_c$ **do**
        Load $\mathbf{K}_j, \mathbf{V}_j \in \mathbb{R}^{B_c \times d}$ from HBM to on-chip SRAM.
        Load $\mathbf{M}_{ij} \in \mathbb{R}^{B_c \times B_c}$ from HBM to on-chip SRAM.
        On chip, compute $\mathbf{A}_{ij} = [\mathbf{Q}_i \mathbf{K}_j^\top] \odot \mathbf{M}_{ij} \in \mathbb{R}^{B_r \times B_c}$.
        On chip, compute $\mathbf{O}_i = \mathbf{O}_i + \mathbf{A}_{ij} \mathbf{V}_j \in \mathbb{R}^{B_r \times d}$.
    **end for**
    Write $\mathbf{O}_i$ to HBM as the $i$-th block of $\mathbf{O}$.
**end for**
return $\mathbf{O}$

---

## C  PROVING ROBUST INFERENCE ALGORITHM

We will use induction to prove: $[\mathbf{kv}]_t = \lambda^{-t}[\overline{\mathbf{kv}}]_t$.

**Base Case** ($n = 1$):

$$\begin{aligned}
[\mathbf{kv}]_1 &= ([\mathbf{kv}]_0 + \mathbf{k_1}\lambda^{-1}\mathbf{v}_1^\top) \\
&= \lambda^{-1}(\mathbf{k_1}\mathbf{v}_1^\top) \\
&= \lambda^{-1}[\overline{\mathbf{kv}}]_1.
\end{aligned} \tag{18}$$

---

**Algorithm 4** Lightning Attention Backward Pass

---

**Input:** $\mathbf{Q}, \mathbf{K}, \mathbf{V}, \mathbf{dO} \in \mathbb{R}^{n \times d}$, attention mask $\mathbf{M} \in \mathbb{R}^{n \times n}$, on-chip SRAM of size $M$, block sizes $B_c, B_r$;
**Initialize:** $\mathbf{dQ} = \mathbf{dK} = \mathbf{dV} = \mathbf{0} \in \mathbb{R}^{n \times d}$;
Divide $\mathbf{Q}$ into $T_r = \frac{n}{B_r}$ blocks $\mathbf{Q}_1, \mathbf{Q}_2, ...\mathbf{Q}_{T_r}$ of size $B_r \times d$ each.
Divide $\mathbf{K}, \mathbf{V}$ into $T_c = \frac{n}{B_c}$ blocks $\mathbf{K}_1, \mathbf{K}_2, ...\mathbf{K}_{T_c}, \mathbf{V}_1, \mathbf{V}_2, ...\mathbf{V}_{T_c}$ of size $B_c \times d$ each.
Divide $\mathbf{O}, \mathbf{dO}$ into $T_r = \frac{n}{B_r}$ blocks $\mathbf{O}_1, \mathbf{O}_2, ...\mathbf{O}_{T_r}, \mathbf{dO_1}, \mathbf{dO_2}, ...\mathbf{dO_{T_r}}$ of size $B_r \times d$ each
Divide $\mathbf{M}$ into $T_r \times T_c$ blocks $\mathbf{M}_{11}, \mathbf{M}_{12}, ...\mathbf{M}_{T_r, T_c}$ of size $B_r \times B_c$ each.
**for** $1 \le j \le T_c$ **do**
    Load $\mathbf{K}_j, \mathbf{V}_j \in \mathbb{R}^{B_c \times d}$ from HBM to on-chip SRAM.
    Initialize $\mathbf{dK}_j = \mathbf{dV}_j = \mathbf{0} \in \mathbb{R}^{B_c \times d}$ on SRAM.
    **for** $1 \le i \le T_r$ **do**
        Load $\mathbf{Q}_i, \mathbf{O}_i, \mathbf{dO}_i \in \mathbb{R}^{B_r \times d}$ from HBM to on-chip SRAM.
        Load $\mathbf{M}_{ij} \in \mathbb{R}^{B_c \times B_c}$ from HBM to on-chip SRAM.
        Initialize $\mathbf{dK}_j = \mathbf{dV}_j = \mathbf{0} \in \mathbb{R}^{B_c \times d}$ on SRAM.
        On chip, compute $\mathbf{A}_{ij} = [\mathbf{Q}_i \mathbf{K}_j^\top] \odot \mathbf{M}_{ij} \in \mathbb{R}^{B_r \times B_c}$.
        On chip, compute $\mathbf{dV}_j = \mathbf{dV}_j + \mathbf{A}_{ij}^\top \mathbf{dO}_i \in \mathbb{R}^{B_c \times d}$.
        On chip, compute $\mathbf{dA}_{ij} = [\mathbf{dO}_i \mathbf{V}_j^\top] \odot \mathbf{M}_{ij} \in \mathbb{R}^{B_r \times B_c}$.
        On chip, compute $\mathbf{dK}_j = \mathbf{dk}_j + \mathbf{dA}_{ij}^\top \mathbf{V}_j \in \mathbb{R}^{B_c \times d}$.
        Load $\mathbf{dQ}_i$ from HBM to SRAM, then on chip, compute $\mathbf{dQ}_i = \mathbf{dK}_i + \mathbf{dA}_{ij} \mathbf{K}_j \in \mathbb{R}^{B_r \times d}$,
        write back to HBM.
    **end for**
    Write $\mathbf{dK}_j, \mathbf{dV}_j$ to HBM as the $j$-th block of $\mathbf{dK}, \mathbf{dV}$.
**end for**
retun $\mathbf{dQ}, \mathbf{dK}, \mathbf{dV}$

---

Assume the statement holds for $n = m - 1$, i.e., $[\mathbf{kv}]_{m-1} = \lambda^{-(m-1)}[\overline{\mathbf{kv}}]_{m-1}$. Then, when $n = m$:

$$
\begin{aligned}
[\mathbf{kv}]_m &= [\mathbf{kv}]_{m-1} + \mathbf{k_m} \lambda^{-m} \mathbf{v}_m^\top \\
&= \lambda^{-(m-1)}[\overline{\mathbf{kv}}]_{m-1} + \mathbf{k_m} \lambda^{-m} \mathbf{v}_m^\top \\
&= \lambda^{-m}(\lambda[\overline{\mathbf{kv}}]_{m-1} + \mathbf{k_m} \mathbf{v}_m^\top) \\
&= \lambda^{-m}[\overline{\mathbf{kv}}]_m,
\end{aligned}
\tag{19}
$$

the statement holds. Therefore, by induction, the statement holds for all $n \ge 1$.

Thus, both the Origin Inference Algorithm and the Robust Inference Algorithm yield the same results.

# D   CORPUS

We gather an extensive corpus of publicly accessible text from the internet, totaling over 700TB in size. The collected data are processed by our data preprocessing procedure as shown in Figure 5, leaving a 6TB cleaned corpus with roughly 2 trillion tokens. We categorize our data sources to provide better transparency and understanding. The specifics of these categories are outlined in Table 11.

## D.1   DATA PREPROCESSING

Our data preprocessing procedure consists of three steps: 1). rule-based filtering, 2). deduplication, and 3). a self-cleaning scheme. Before being added to the training corpus, the cleaned corpus needs to be evaluated by humans.

**Rule-based filtering**    The rules we used to filter our collected data are listed as follows:

- *Removal of HTML Tags and URLs:* The initial step in our process is the elimination of HTML tags and web URLs from the text. This is achieved through regular expression techniques that identify these patterns and remove them, ensuring the language model focuses on meaningful textual content.
- *Elimination of Useless or Abnormal Strings:* Subsequently, the cleaned dataset undergoes a second layer of refinement where strings that do not provide value, such as aberrant strings

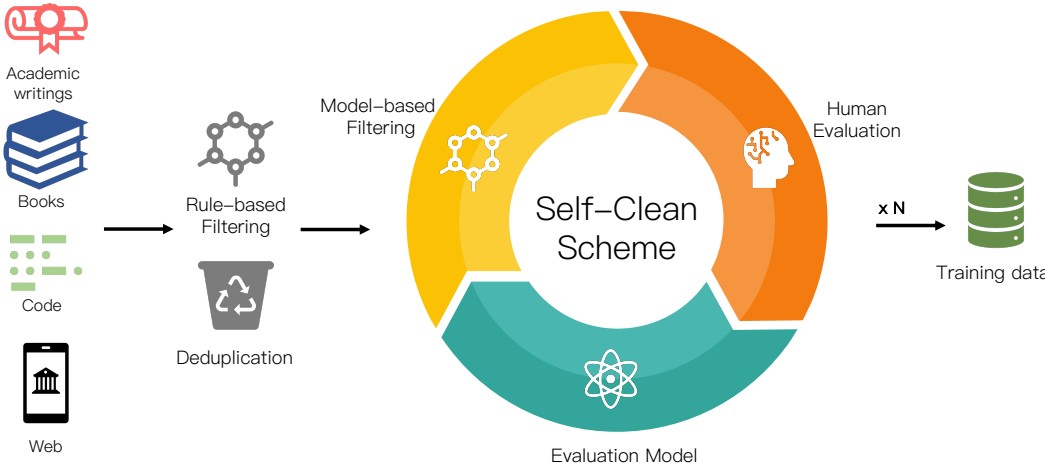

Figure 5: **Data Preprocess Procedure.** The collected data undergoes a process of rule-based filtering and deduplication, followed by our self-clean data processing strategy: model-based filtering, human evaluation, and evaluation model. After several iterations of the above cycle, we obtain high-quality training data at around 2T tokens.

or garbled text, are identified and excised. This process relies on predefined rules that categorize certain string patterns as non-contributing elements.

- *Deduplication of Punctuation Marks:* We address the problem of redundant punctuation marks in the data. Multiple consecutive punctuation marks can distort the natural flow and structure of sentences when training the model. We employ a rule-based system that trims these duplications down to a single instance of each punctuation mark.
- *Handling Special Characters:* Unusual or special characters that are not commonly part of the language's text corpus are identified and either removed or replaced with a standardized representation.
- *Number Standardization:* Numerical figures may be presented in various formats across different texts. These numbers are standardized into a common format to maintain consistency.
- *Preservation of Markdown/LaTeX Formats:* While removing non-textual elements, exceptions are made for texts in Markdown and LaTeX formats. Given their structured nature and ubiquitous use in academia and documentation, preserving these formats can enhance the model's ability to understand and generate similarly formatted text.

**Deduplication** To ensure the uniqueness of our data and avert the risk of overfitting, we employ an efficient de-duplication strategy at the document or line level using MinHash and Locality-Sensitive Hashing (LSH) algorithms. This combination of MinHash and LSH ensures a balance between computational efficiency and accuracy in the deduplication process, providing a robust mechanism for data deduplication and text watermark removal.

**Self-cleaning scheme** Our data self-cleaning process involves an iterative loop of the following three steps to continuously refine and enhance the quality of our dataset. An issue of using model-based data filters is that the filtered data will have a similar distribution as the evaluation model, which may have a significant impact on the diversity of the training data. Assuming that the majority of the pre-processed data is of high quality, we can train an evaluation model on the entire set of pre-processed data, and the model will automatically smooth the data manifold distribution and outlet low-quality data while retaining the majority of the diversities.

The self-cleaning scheme unfolds as follows:

- *Evaluation Model:* We train a 385M model on the pre-processed corpus to act as a data quality filter.

Table 11: **Statistics of our corpus.** For each category, we list the number of epochs performed on the subset when training on the 2 trillion tokens, as well as the number of tokens and disk sizes. We also list the table on the right according to the language distribution.

| Dataset | Epochs | Tokens | Disk size |
|---|---|---|---|
| Academic Writings | 1.53 | 200 B | 672 GB |
| Books | 2.49 | 198 B | 723 GB |
| Code | 0.44 | 689 B | 1.4 TB |
| Encyclopedia | 1.51 | 5 B | 18 GB |
| Filtered Webpages | 1.00 | 882 B | 3.1 TB |
| Others | 0.63 | 52 B | 154 GB |
| Total | - | 2026 B | 6 TB |

| Language | Tokens | Disk size |
|---|---|---|
| English | 743 B | 2.9 TB |
| Chiese | 555 B | 1.7 TB |
| Code | 689 B | 1.4 TB |
| Others | 39 B | 89 GB |
| Total | 2026 B | 6 TB |

- *Model-Based Data Filtering:* We use the evaluation model to assess each piece of data with perplexity. Only data achieving a score above a certain threshold is preserved for the next step. Low-quality data are weeded out at this stage.
- *Human Evaluation:* We sample a small portion of the filtered data and manually evaluate the quality.

These steps are repeated in cycles, with each iteration improving the overall quality of the data and ensuring the resulting model is trained on relevant, high-quality text. This self-cleaning process provides a robust mechanism for maintaining data integrity, thereby enhancing the performance of the resulting language model.

## D.2 TOKENIZATION

We tokenize the data with the Byte-Pair Encoding (BPE) algorithm. Notably, to enhance compatibility with Chinese language content, a significant number of common and uncommon Chinese characters have been incorporated into our vocabulary. In cases where vocabulary items are not present in the dictionary, the words are broken down into their constituent UTF-8 characters. This strategy ensures comprehensive coverage and flexibility for diverse linguistic input during model training.

## E ADDITIONAL EXPERIMENTAL RESULTS

### E.1 MODEL PARALLELISM ON TRANSNORMERLLM

We conduct a series of experiments with a 7B TransNormerLLM model to investigate the performance of model parallelism on TransNormerLLM in terms of speed and memory. These tests are carried out on a single Nvidia DGX node that houses eight A100 80G GPUs linked by NVLink. In this experiment, FSDP is enabled and Flash Attention (Dao et al., 2022a) is used on the Transformer. Table 12 shows the results for training speed and memory consumption.

It can be seen that model parallelism has a significant effect on memory conservation, as increasing the number of partitions for the model results in lower memory consumption per GPU. Due to NVLink constraints, we kept the dimension of model parallelism within 8 in all of our experiments. The TransNormerLLM-7B model requires only 24.1GB of memory on a single GPU when the model parallel size is set to 8, representing a significant memory reduction of 62.3% when compared to the model parallel size of 1. In comparison, the Transformer-7B model consumes 28.7GB of memory under the same configuration. While model parallelism conserves memory, it is worth noting that training speed is only marginally reduced. TransNormerLLM consistently outperforms Transformer by a wide margin.

### E.2 STRESS TESTS ON MODEL SIZE AND CONTEXT LENGTH

A series of stress tests are performed to assess the efficacy of the designed system optimization strategy. The model is scaled up to 175B, which is the largest released version of the TransNormerLLM model. However, this augmentation poses significant training challenges. We use a wide range of distributed training techniques to effectively train such a large model, with the goal of reducing GPU memory consumption while increasing computational and communication efficiencies. To ensure the

Table 12: **Model Parallelism Performance.** We compare the model parallelism performance of Transformer-7B with Flash Attention and TransNormerLLM-7B with Lightning Attention on a single A100 node with NVLink. All experiments use a batch size of 2 and a context length of 2048.

| Model | Model Parallel Size | Tokens/s | Allocated Memory/GPU | Memory Saved |
|---|---|---|---|---|
| | 1 | 26896.1 | 66.3 GB | - |
| Transformer-7B | 2 | 24973.7 | 44.6 GB | 32.7% |
| | 4 | 22375.8 | 40.2 GB | 39.4% |
| | 8 | 19973.6 | 28.7 GB | 56.7% |
| | 1 | 32048.6 | 64.0 GB | - |
| TransNormerLLM-7B | 2 | 29750.4 | 41.0 GB | 35.9% |
| | 4 | 27885.2 | 36.3 GB | 43.3% |
| | 8 | 24280.0 | 24.1 GB | 62.3% |

feasibility of training these massive TransNormerLLM models, Lightning Attention, FSDP, Model Parallelism, AMP, and Activation Checkpointing are used. For the Transformer models, we use Flash Attention (Dao et al., 2022a) in all experiments.

**Model Size**    We perform training experiments on variously sized Transformer and TransNormer-LLM models using a large-scale A100 80G GPU cluster, as shown in Table 13. To achieve the maximum speed for various model sizes, we keep the context length constant at 2048 and increased the batch size until we reached the GPU memory limit. TransNormerLLMs consistently outperform their Transformer counterparts in terms of computation speed. This observation validates the TransNormerLLM model's advantageous linear computational complexity, reinforcing its efficacy.

Table 13: **Efficiency of training models with different sizes.** For comparative purposes, we keep the context length fixed at 2048 and increased the batch size for both transformer and TransNormerLLM to achieve their maximum speeds without encountering out-of-memory issues.

| Model | Model Size | Tokens/sec/GPU | Allocated Memory/GPU |
|---|---|---|---|
| | 7B | 3362.7 | 72.5 GB |
| Transformer | 13B | 1735.6 | 70.6 GB |
| | 65B | 318.2 | 73.2 GB |
| | 175B | 106.2 | 69.5 GB |
| | 7B | 4081.0 | 71.9 GB |
| TransNormerLLM | 13B | 2104.3 | 73.8 GB |
| | 65B | 406.9 | 69.4 GB |
| | 175B | 136.6 | 70.3 GB |

**Context Length**    One of the strengths of TransNormerLLM lies in its utilization of linear attention computation, which exhibits computational and storage complexities linearly correlated with the sequence length. To validate this outstanding characteristic of TransNormerLLM, we conduct training experiments on Transformer and TransNormerLLM models with varying parameter sizes. While maintaining a batch size of 1, we aim to maximize the context length. All experiments run on a small cluster with 64 A100 GPUs. The results, as presented in Table 14, demonstrate the remarkable long context length training capability of TransNormerLLM. Under comparable computational resources, the TransNormerLLM model exhibits the ability to train with longer context lengths compared to conventional Transformer models and achieve higher computational speeds in the process.

Table 14: **Maximum context length for training Transformer and TransNormerLLM.** We compare the maximum context lengths with different model sizes between Transformer and TransNormer-LLM on 64 A100 80G GPUs. All experiments use a batch size of 1.

| Model | Model Size | Context Length | Relative Speed | Allocated Memory/GPU |
|---|---|---|---|---|
| Transformer | 7B | 37K | 1 | 71.1 GB |
| | 13B | 24K | 1 | 68.0 GB |
| | 65B | 19K | 1 | 73.3 GB |
| | 175B | 10K | 1 | 66.9 GB |
| TransNormerLLM | 7B | 48K | 1.21 | 65.8 GB |
| | 13B | 35K | 1.23 | 61.0 GB |
| | 65B | 23K | 1.29 | 68.2 GB |
| | 175B | 12K | 1.35 | 63.5 GB |

