# OpenReview forum: "TransNormerLLM: A Faster and Better Large Language Model with Improved TransNormer"
_ICLR.cc/2024/Conference — Submitted to ICLR 2024_

### Official Review · Reviewer_5syF · 2023-10-26

**Soundness:** 2 fair
**Presentation:** 3 good
**Contribution:** 2 fair
**Rating:** 5
**Confidence:** 4

**Summary:**

This paper presents TransNormerLLM and claims the first linear attention-based LLM that outperforms conventional softmax-based models in terms of both accuracy and efficiency.

The adopted techniques include (1) LRPE position encoding together with an exponential decay to avoid attention dilution issues; (2) a gating mechanism following Flash; and (3) Lightning Attention to accelerate the training process.

Experiments on various scale LLM models trained on self-collected corpus demonstrate outstanding performance.

**Strengths:**

1. The pioneering work focusing on the linear attention-based LLM. This direction is of great interest and significance to the community.

2. Combine various tricks to make linearized LLM work, including the aforementioned LRPE position encoding, gated linear attention or MLPs, and lightning attention.

3. Through evaluation and ablation studies to reveal the core techniques that make linear attention work.

**Weaknesses:**

1. How does your linear attention handle the autoregressive decoding? As of training, you can feed the network with a batch of inputs with long token dimensions. But when it comes to the generation phase, I am afraid that only limited tokens are used to generate the next token. Then do you still have benefits for inference?

2. The paper reads like a combination of various tricks as a lot of techniques were discussed in the previous paper, like LRPE, Flash, and Flash Attention. Especially for the Lightning Attention vs. Flash Attention, I did not find any difference between these two. The gated mechanism was also introduced in Flash paper. These aspects leave us a question in terms of the technical novelty of this paper.

3. It looks like during training, you are still using the quadratic attention computational order as indicated in Equ. 10? I suppose it was to handle the masking part. But that loses the benefits of training with linear attention complexity.

4. In terms of evaluation, although in the abstract, the authors claim that the linearized LLM extends to 175B parameters, most experiments are conducted on 375M models. For the large parameter size settings, the author only reports the memory and latency cost savings. The accuracy information is missing, without which I feel hard to evaluate the linearized LLMs.

**Questions:**

See weaknesses.

---

> ### Author Response · Authors · 2023-11-21
> **Response to Reviewer 5syF(Part 1)**
>
> Q1. How does attention handle autoregressive decoding?
>
> Inference can be divided into two stages. The first stage involves calculating the key-value (kv) based on the input prompt. This is done using the following code:
>
> ```
> # b, h, n, e, d
> kv_outproduct = torch.einsum("... n e, ... n d -> ... n e d", k, v)
> # 1, 1, n, 1, 1
> index = torch.arange(n - 1, -1, -1).reshape(1, 1, -1, 1, 1).to(x)
> decay = ratio.unsqueeze(0).unsqueeze(-1) ** index
>
> kv_outproduct_with_decay = kv_outproduct * decay
> kv = torch.sum(kv_outproduct_with_decay, dim=-3)
> ```
>
> The second stage involves using the calculated kv and computing the output. This is done using the following code:
>
> ```
> kv = ratio * kv + torch.einsum(
>     "... n d, ... n e -> ... d e",
>     k,
>     v,
> )
> output = torch.einsum(
>     "... n e, ... e d -> ... n d", q, kv
> )
> ```
>
> Please note that Stage 1 is executed only once, and the cost of each inference is mostly determined by Stage 2. The time and space complexity for Stage 2 are both $O(bhd)$, where b represents the batch size, h represents the number of heads, and d represents the head dimension, which is usually either 64 or 128.
>
> We provide a speed comparison in the table below for different sequence lengths, measured in tokens/second/gpu. The larger the value, the better:
>
> | Model/Seqlen | 512 | 1024 | 2048 | 4096 | 8192 |
> | --- | --- | --- | --- | --- | --- |
> | Transformer | 133.44 | 86.69 | 34.95 | 12.42 | 3.69 |
> | TransnormerLLm | 274.48 | 158.41 | 86.52 | 43.97 | 22.88 |
>
> Q2. The contribution of this work.
>
> Scaling up a linear attention-based language model and making it faster and better than transformer-based models is a non-trivial task. It requires extensive optimization of the whole system rather than a small part of the model. As shown in this paper, we make crucial enhancements to nearly every part of the model, such as the architecture, the attention mechanism, the inference algorithm, and the model parallel implementation.
>
> Regarding the difference between lightning attention and flash attention, our lightning attention is IO-friendly linear attention, while the flash attention is IO-friendly softmax attention. We agree that Flash Attention sheds light on our lightning attention. By incorporating this idea, we have successfully addressed the issue of slow speed in the causal scenario of Linear Attention mentioned in "Flash: Transformer Quality in Linear Time". In addition, by leveraging the characteristics of Linear Attention, our lightning attention can achieve significant efficiency improvements compared to Flash 1/2. We provide a comparison of the efficiency between our method and Flash Attention 1/2 in the following section.
>
> Q3. The computing method of lighting attention.
>
> In our preliminary version, we use eq 10 implementations to avoid the slow looping issue mentioned in "Flash: Transformer Quality in Linear Time". However, we have further optimized our approach by leveraging the mechanisms of linear attention, resulting in enhanced efficiency. Through these optimizations, our lightning attention demonstrates significant improvements in efficiency compared to Flash 1/2. Below, we provide an efficiency comparison between our method and Flash Attention 1/2. The speed table is measured in seconds, while the memory table is measured in MB. Smaller values indicate superior performance in both speed and memory.
>
> Speed table(seconds):
>
> Forward:
>
> | Seqlen | Lightning | Flash2 | Flash1 |
> | --- | --- | --- | --- |
> | 1024.0 | 0.255829 | 0.188746 | 0.282317 |
> | 2048.0 | 0.408914 | 0.459194 | 0.875625 |
> | 4096.0 | 0.808699 | 1.552662 | 3.094903 |
> | 8192.0 | 1.601662 | 5.735785 | 11.579776 |
> | 16384.0 | 3.197140 | 22.200319 | 44.999168 |
> | 32768.0 | 6.380953 | 86.720512 | 178.348038 |
>
> Backward:
>
> | Seqlen | Lightning | Flash2 | Flash1 |
> | --- | --- | --- | --- |
> | 1024.0 | 0.647602 | 0.526302 | 0.786245 |
> | 2048.0 | 1.268677 | 1.516766 | 2.427769 |
> | 4096.0 | 2.507520 | 4.948076 | 8.316556 |
> | 8192.0 | 4.990168 | 17.650484 | 30.684160 |
> | 16384.0 | 9.925859 | 66.620415 | 117.788673 |
> | 32768.0 | 19.840511 | 258.737152 | 461.466614 |
>
> Memory Table(MB):
>
> Forward:
> | Seqlen | Lightning | Flash2 | Flash1 |
> | --- | --- | --- | --- |
> | 1024.0 | 64.000488 | 80.500488 | 96.500977 |
> | 2048.0 | 128.000488 | 161.000488 | 193.000977 |
> | 4096.0 | 256.000488 | 322.000488 | 386.000977 |
> | 8192.0 | 512.000488 | 644.000488 | 772.000977 |
> | 16384.0 | 1024.000488 | 1288.000488 | 1544.000977 |
> | 32768.0 | 2048.000488 | 2576.000488 | 3088.000977 |
>
> Backward:
> | Seqlen | Lightning | Flash2 | Flash1 |
> | --- | --- | --- | --- |
> | 1024.0 | 166.400488 | 215.400488 | 215.400488 |
> | 2048.0 | 332.800488 | 430.800488 | 430.800488 |
> | 4096.0 | 665.600488 | 861.600488 | 861.600488 |
> | 8192.0 | 1331.200488 | 1723.200488 | 1723.200488 |
> | 16384.0 | 2662.400488 | 3446.400488 | 3446.400488 |
> | 32768.0 | 5324.800488 | 6892.800488 | 6892.800488 |

---

> > ### Author Response · Authors · 2023-11-21
> > **Response to Reviewer 5syF(Part 2)**
> >
> > Q4. The evaluation of larger models.
> >
> > Thank you for your comments. We understand your concerns about the evaluation of our model, especially for large-scale settings. Due to the computational intensity, ablation studies could only be conducted on smaller-scale models. Since we are using the same setting and corpus in our ablation studies, the conclusion is still valid for larger models. While most of the ablation studies for architecture design are performed on the 385M model, we also provide benchmark results for 1B and 7B models. As a preliminary study on scaling linear transformers, we have systematically proven that our transNormerLLM can achieve faster training and lower memory consumption than transformer-based models in the 175B setting. It is worth noting that scaling up to 175B is nontrivial, and it requires a sophisticated model parallel implementation while existing model parallel techniques are only suitable for transformer structures. For the purpose of accuracy validation, we benchmarked our model with a 7B size, which is a frequently used size for existing transformer-based LLMs. We appreciate your patience and understanding as completing validation of the full 175B model will take a significant amount of time. However, we aim to include these results in future work when they become available.

---

### Official Review · Reviewer_s89T · 2023-11-02

**Soundness:** 3 good
**Presentation:** 2 fair
**Contribution:** 3 good
**Rating:** 6
**Confidence:** 4

**Summary:**

This paper proposes several incremental changes upon TransNormer including: positional embedding, gated linear units and layernorm. The authors claim that the resulting model is not only more effective, but also more efficient during both training and inference, compared with the original TransNormer.

**Strengths:**

- Extensive empirical results with detailed ablation on the proposed changes
- Detailed evaluation of both training and inference

**Weaknesses:**

- changes are incremental, thus the work lack technical contribution
- the training and inference algorithms are not novel (explained more blow)
- the results are not convincing due to baseline setup and implementations

Training efficiency: From the appendix, it seems the implementation of the linear attention is a direct adaptation of Flash-Attention2 (I further assume it’s based on the Triton implementation of Flash-Attention2). It’s unfair (and not meaningful) to compare the flash-attention optimized Transnormer with a plain Pytorch Transformer.

Baseline: for fair comparison, the baseline Transformer should also positional embeddings or gated linear units (which are straightforward to be added to Transformer as well).

Robust inference: the proposed new inference algorithm is rather trivial. It’s more natural (and conventional in the literature) to include the decay factor in the recurrence (i.e., $ h_t = \lambda \  h_{t-1} + k_t v_t^\intercal $ ) rather than in the input and output.

**Questions:**

Can you compare against RetNet[1] in Sec 4.2? Transnormer is quite similar to RetNet with the very similar design of decay and positional embeddings. It'd be useful for authors to clarify the difference and make empirical comparisons.

1. Retentive Network: A Successor to Transformer for Large Language Models

---

> ### Author Response · Authors · 2023-11-21
> **Response to Reviewer s89T**
>
> Q1. The contribution of TransnormerLLM.
>
> The technical contribution of this paper is to build a linear attention-based large language model with faster speed and better performance than transformer-based ones. Prior to our work, no previously developed language models had achieved the same level of effectiveness as our model. Our approach involved extensive ablations to validate the network architecture, and significant engineering efforts to accelerate linear attention and scale up the model using advanced model parallel implementation. As a result, our model achieved unprecedented levels of performance, surpassing that of any prior efficient language models. We have pushed the boundaries of what linear attention can achieve in terms of model size, which we believe represents a significant advancement in the field.
>
> Q2. Lightning attention algorithm, training efficiency, and baseline.
>
> We understand your concerns about the fairness of comparison. However, it's worth noting that our comparison object is an optimized Flash attention Transformer, not a plain PyTorch Transformer. The comparison aims to demonstrate the improvements achieved by our approach over advanced models like Flash Attention Transformer in terms of speed and memory management. Detailed comparisons can be found in Tables 12 and 13 in our paper. We further optimized the algorithm and compared its speed and memory usage with Flash2 and Flash1. The units in the speed table are seconds, while the units in the memory table are in MB. Smaller values indicate better performance in both speed and memory.
>
> Forward:
>
> | Seqlen | Lightning | Flash2 | Flash1 |
> | --- | --- | --- | --- |
> | 1024.0 | 0.255829 | 0.188746 | 0.282317 |
> | 2048.0 | 0.408914 | 0.459194 | 0.875625 |
> | 4096.0 | 0.808699 | 1.552662 | 3.094903 |
> | 8192.0 | 1.601662 | 5.735785 | 11.579776 |
> | 16384.0 | 3.197140 | 22.200319 | 44.999168 |
> | 32768.0 | 6.380953 | 86.720512 | 178.348038 |
>
> Backward:
>
> | Seqlen | Lightning | Flash2 | Flash1 |
> | --- | --- | --- | --- |
> | 1024.0 | 0.647602 | 0.526302 | 0.786245 |
> | 2048.0 | 1.268677 | 1.516766 | 2.427769 |
> | 4096.0 | 2.507520 | 4.948076 | 8.316556 |
> | 8192.0 | 4.990168 | 17.650484 | 30.684160 |
> | 16384.0 | 9.925859 | 66.620415 | 117.788673 |
> | 32768.0 | 19.840511 | 258.737152 | 461.466614 |
>
> Memory Table(MB):
>
> Forward:
>
> | Seqlen | Lightning | Flash2 | Flash1 |
> | --- | --- | --- | --- |
> | 1024.0 | 64.000488 | 80.500488 | 96.500977 |
> | 2048.0 | 128.000488 | 161.000488 | 193.000977 |
> | 4096.0 | 256.000488 | 322.000488 | 386.000977 |
> | 8192.0 | 512.000488 | 644.000488 | 772.000977 |
> | 16384.0 | 1024.000488 | 1288.000488 | 1544.000977 |
> | 32768.0 | 2048.000488 | 2576.000488 | 3088.000977 |
>
> Backward:
> | Seqlen | Lightning | Flash2 | Flash1 |
> | --- | --- | --- | --- |
> | 1024.0 | 166.400488 | 215.400488 | 215.400488 |
> | 2048.0 | 332.800488 | 430.800488 | 430.800488 |
> | 4096.0 | 665.600488 | 861.600488 | 861.600488 |
> | 8192.0 | 1331.200488 | 1723.200488 | 1723.200488 |
> | 16384.0 | 2662.400488 | 3446.400488 | 3446.400488 |
> | 32768.0 | 5324.800488 | 6892.800488 | 6892.800488 |
>
> Q3. The robust inference algorithm is trivial.
>
> It is noteworthy that the issue of robust inference in linear transformers has not been explored previously in this field. The body of literature on linear attention with exponential decay is somewhat limited, and to date, no studies have been conducted on its corresponding inference. In this paper, we first reveal the existence of the problem and then provide a mathematically rigorous solution for it. We believe that our contribution to the growth of large language models is valuable.
>
> Q4. Experiments result about RetNet.
>
> It is noteworthy that our work and the RetNet paper were conducted simultaneously, which made it impossible for us to make a direct comparison during the research process. We also noticed that RetNet has been submitted to this ICLR conference, as seen in the [link](https://openreview.net/forum?id=UU9Icwbhin). Nevertheless, we evaluated one of their models to draw a comparison, and the results are presented in the following table. Our model performed significantly better than RetNet.
>
> | model | params | loss |
> | --- | --- | --- |
> | transnormerllm | 385M | 2.248 |
> | RetNet | 391M | 2.371 |

---

> > ### Comment · Reviewer_s89T · 2023-11-22
> >
> > Thanks for the clarifications!
> >
> > * Can you elaborate more on "We further optimized the algorithm and compared its speed and memory usage with Flash2 and Flash1."
> >
> > * I agree it's not necessary to compare against RetNet, can you clarify why it can achieve better performance than RetNet in your new experiments?
> >
> > Thanks.

---

> > > ### Author Response · Authors · 2023-11-23
> > > **Response to Reviewer s89T**
> > >
> > > Q1. More details about algorithm optimization
> > >
> > > In our implementation, we utilized the right product of linear attention. We successfully solved the slow loop issue mentioned in "Flash: Transformer Quality in Linear Time" with extensive engineering optimization, which enabled our algorithm to be faster than Flash 2. We will release our code for the open-source community in the near furture.
> > >
> > > Q2. The reason for better performance than RetNet.
> > >
> > > The main differences between TransNormerLLM and RetNet are as follows:
> > >
> > > 1. Different values are chosen for exp decay.
> > > 2. TransNormerLLM uses a learnable lrpe (learnable relative positional encoding), while RetNet uses xPos (absolute positional encoding).
> > > 3. TransNormerLLM utilizes GLU (gated linear unit), while RetNet uses FFN (feed-forward network).
> > > 4. The number of parameters in the linear attention part is $5d^2$ in TransNormerLLM, and $8d^2$ in the GLU part. In RetNet, the number of parameters in the linear attention part is $8d^2$, and in the FFN part is $4d^2$.
> > >
> > > In summary, there are several specific network design details that differ between the two models. This experiment demonstrates that the design of TransNormerLLM is superior to that of RetNet.

---

### Official Review · Reviewer_xvcu · 2023-11-02

**Soundness:** 3 good
**Presentation:** 3 good
**Contribution:** 2 fair
**Rating:** 5
**Confidence:** 5

**Summary:**

The paper proposes TransNormerLLM, an improved linear attention-based large language model (LLM) that outperforms conventional softmax attention models in both accuracy and efficiency.

- The key contributions include:

1. Replacing the TransNormer's DiagAttention with Linear Attention and using LRPE (Learnable Relative Positional Encoding) with exponential decay to allow global token interactions while avoiding attention dilution.

2. Introducing Lightning Attention to accelerate linear attention by 2x during training and reduce memory usage by 4x.

3. Using a gating mechanism for training stability and simplified GLU/normalization for faster processing.

4. Developing a robust inference algorithm for consistent speed regardless of sequence length.

5. Implementing efficient model parallelism to scale the model up to 175B parameters.

6. Validating the model on a 6TB corpus and benchmarking 385M, 1B and 7B parameter models, showing competitive performance to Transformer LLMs while being faster.

**Strengths:**

- The work addresses a key limitation of standard Transformer models - the quadratic complexity w.r.t sequence length - through the use of linear attention. This could enable scaling to much longer contexts.

- The modifications to the original TransNormer architecture, especially Lightning Attention and robust inference, significantly improve efficiency and stability.

- Thorough ablation studies validate the impact of each proposed technique. The benchmarking shows the models match state-of-the-art Transformer LLMs in accuracy while being faster.

- The model parallelism enables scaling up to 175B parameters, allowing large-scale pre-training. The efficiency gains are impressive and impactful.

- The code and models will be open-sourced, promoting further research and application of efficient transformers.

**Weaknesses:**

- The novelty of the work is fairly incremental, building directly on prior work like TransNormer and Flash Attention. None of the modifications substantially advance the state-of-the-art.

- While the techniques improve efficiency, the gains in accuracy over standard TransNormer appear marginal based on the results. The ablations also suggest the contributions are optimizations rather than modelling improvements.

- Lightning Attention, while providing speed and memory gains, is not particularly novel, simply applying similar ideas from Flash Attention.

- The robust inference modification is motivated through mathematical derivations, but the practical benefits are unclear. There is no evidence it improves stability or accuracy.

- The pre-training data is underspecified, making fair comparison to other models difficult.

**Questions:**

- What is the actual impact of the robust inference modification in practice? Any quantitative results?

- Can the authors provide more details on the pre-training data characteristics and compute resources used? Is there any testing data leakage problem?

- The gains over standard Transformers appear quite large but the gains over TransNormer are marginal. Why is this the case? Can you show the training curves for the models?

---

> ### Author Response · Authors · 2023-11-20
> **Response to Reviewer xvcu(Part1)**
>
> Q1. The contribution of this work.
>
> Our TransNormerLLM is built on TransNormer and Flash Attention, with crucial enhancements, including architecture optimization, algorithm adaptation and optimization, and model parallel implementation. The novelty of this work lies in how to build a linear attention-based language model with faster speed and better performance. Prior to our work, none of the previously developed efficient language models, including those based on linear attention, sparse attention, RNNs, and LongConv, had achieved the same level of effectiveness as our model. By combining algorithmic developments with engineering efforts, we have pushed the boundaries of what linear attention is achievable regarding model size, which we believe constitutes a substantial advancement in the field.
>
> Q2. The motivation of this work and the advantages compared to TransNormer.
>
> Our work focuses on improving efficiency rather than accuracy compared to the standard TransNormer model to enable successful scaling. As we have claimed in the abstract: ”TransNormerLLM evolves from the previous linear attention architecture TransNormer by making advanced modifications”. The key contribution of this work is scaling linear attention to large language models with significant efficiency improvement and comparable performances to transformer-based LLM models. Note that TransNormerLLM also requires less memory compared to TransNormer-T1/T2:
> | model | memory |
> | --- | --- |
> | transnormerllm | 29.3 |
> | transnormer_t1 | 34.6 |
> | transnormer_t2 | 33.8 |
>
> Besides, the modeling differences between TransnormerLLM and Transnormer are extensive including position encoding, token mixing methods, etc. In addition, the optimization such as model parallelism, and lightning attention are none trivial and they require significant engineering efforts. These modifications made up our key contributions.
>
> Q3. The motivation and significance of Lightning Attention.
>
> We agree that Flash Attention sheds light on our lightning attention. By incorporating this idea, we have successfully addressed the issue of slow speed in the causal scenario of Linear Attention mentioned in "Flash: Transformer Quality in Linear Time". Additionally, we have further optimized our approach by leveraging the mechanisms of linear attention, resulting in enhanced efficiency. Through these optimizations, our lightning attention demonstrates significant improvements in efficiency compared to Flash 1/2. Below, we provide an efficiency comparison between our method and Flash Attention 1/2. The speed table is measured in seconds, while the memory table is measured in MB. Smaller values indicate superior performance in both speed and memory.
>
> Speed table(seconds):
>
> Forward:
>
> | Seqlen | Lightning | Flash2 | Flash1 |
> | --- | --- | --- | --- |
> | 1024.0 | 0.255829 | 0.188746 | 0.282317 |
> | 2048.0 | 0.408914 | 0.459194 | 0.875625 |
> | 4096.0 | 0.808699 | 1.552662 | 3.094903 |
> | 8192.0 | 1.601662 | 5.735785 | 11.579776 |
> | 16384.0 | 3.197140 | 22.200319 | 44.999168 |
> | 32768.0 | 6.380953 | 86.720512 | 178.348038 |
>
> Backward:
>
> | Seqlen | Lightning | Flash2 | Flash1 |
> | --- | --- | --- | --- |
> | 1024.0 | 0.647602 | 0.526302 | 0.786245 |
> | 2048.0 | 1.268677 | 1.516766 | 2.427769 |
> | 4096.0 | 2.507520 | 4.948076 | 8.316556 |
> | 8192.0 | 4.990168 | 17.650484 | 30.684160 |
> | 16384.0 | 9.925859 | 66.620415 | 117.788673 |
> | 32768.0 | 19.840511 | 258.737152 | 461.466614 |
>
> Memory Table(MB):
>
> Forward:
>
> | Seqlen | Lightning | Flash2 | Flash1 |
> | --- | --- | --- | --- |
> | 1024.0 | 64.000488 | 80.500488 | 96.500977 |
> | 2048.0 | 128.000488 | 161.000488 | 193.000977 |
> | 4096.0 | 256.000488 | 322.000488 | 386.000977 |
> | 8192.0 | 512.000488 | 644.000488 | 772.000977 |
> | 16384.0 | 1024.000488 | 1288.000488 | 1544.000977 |
> | 32768.0 | 2048.000488 | 2576.000488 | 3088.000977 |
>
> Backward:
> | Seqlen | Lightning | Flash2 | Flash1 |
> | --- | --- | --- | --- |
> | 1024.0 | 166.400488 | 215.400488 | 215.400488 |
> | 2048.0 | 332.800488 | 430.800488 | 430.800488 |
> | 4096.0 | 665.600488 | 861.600488 | 861.600488 |
> | 8192.0 | 1331.200488 | 1723.200488 | 1723.200488 |
> | 16384.0 | 2662.400488 | 3446.400488 | 3446.400488 |
> | 32768.0 | 5324.800488 | 6892.800488 | 6892.800488 |
>
> Q4. The robust inference algorithm.
>
> In our work, the robust inference modification is indeed mathematically motivated, and they are completely equivalent in terms of theory. We have tested the numerical errors of Origin Inference and Robust Inference, as shown in the following table. The parameters used for the test were b=2, h=8, n=1024, and d=64, where b represents the batch size, h represents the number of heads, n represents the sequence length, and d represents the feature dimension. It can be observed that the robust inference algorithm does not have numerical precision issues.
>  | Origin | Robust |
>  | --- | --- |
>  | nan | 0.0238 |

---

> ### Author Response · Authors · 2023-11-20
> **Response to Reviewer xvcu(Part 2)**
>
> Q5. The pre-training dataset.
>
> We understand your concern about the specificity of the pre-training data, which can indeed affect a fair comparison with other models. However, it is common practice for LLM not to open-source pre-training data due to various reasons including privacy and security concerns. For instance, Llama1/2 do not disclose their pre-training data, thus it is impossible for us to obtain their training data information. Our evaluation protocol for LLMs adheres to standard practices, where we perform assessments via downstream tasks. To ensure fairness in model design comparison, all the ablation studies mentioned in our paper were conducted using the same dataset, ensuring internal consistency and validity of the conclusions drawn.
>
> Q6. The details on the pre-training data characteristics and compute resources used.
>
>  For more detailed information about the pre-training data characteristics and compute resources used, please refer to Appendix D of our paper where we have provided a comprehensive description. To address your concern about potential data leakage, we've taken strict measures during data selection to ensure there's no occurrence of such issues. We understand the importance of maintaining the integrity of our experiments and ensuring that our results are accurately reflective of the model's performance.
>
> Q7. The learning curve.
>
> It is important to mention that TransNormer has surpassed Transformer in language modeling, as shown in table 4 of TransNormer paper. While TransNormerLLM primarily aims to enhance efficiency, it also yields a positive impact on performance.
>
> As for the training curves, we present the training loss in the form of a table rather than a curve due to space constraints. However, we believe that the table still provides a clear visualization of the training process and performance improvements over time.
>
> | model/steps | 20k | 40k | 60k | 80k | 100k |
> | --- | --- | --- | --- | --- | --- |
> | transnormerllm | 2.563 | 2.415 | 2.352 | 2.298 | 2.248 |
> | transnormer_t1 | 2.581 | 2.423 | 2.378 | 2.316 | 2.29 |
> | transnormer_t2 | 2.571 | 2.426 | 2.363 | 2.311 | 2.274 |
> | transformer | 2.987 | 2.621 | 2.482 | 2.401 | 2.362 |

---

### Official Review · Reviewer_LFwx · 2023-11-06

**Soundness:** 4 excellent
**Presentation:** 4 excellent
**Contribution:** 4 excellent
**Rating:** 8
**Confidence:** 4

**Summary:**

This work proposes the TransNormerLLM model that builds on recent advancements in linear attention model techniques such as TransNormer with gating as model architecture, LRPE with exponential decay for position encoding. This work also speeds up causal Linear attention computation to make it io-aware.  Furthermore, model parallelism is introduced for the SGLU and GLA blocks for efficient large-scale distribute training. The architectural and efficient training techniques lead to faster training while matching the performance of the Transformer architecture. Finally, a robust inference for exponential decay attention is presented for numerical stability.

**Strengths:**

- The paper is well-written clearly explaining the contributions and motivations.
- The experiments are well-conducted, with thoughtful ablations and comparisons to extensive range of baselines. This is an important advancement showing that linear attention can match the Transformer performance at scale. Overall I think this work will help to significantly reduce the computation and better scale LLMs.
- Detailed information regarding the corpus, pseudo-codes, model hyper-parameters is provided,  aiding reproducibility.

**Weaknesses:**

- One suggestion would be to explicitly state what is meant by exponential decay position encoding in section 3.1.1.  My understanding is in this case $a_{st} = q_s^T k_t \lambda^{(s-t)}$. Please correct me if I am wrong.
- While the training efficiency is provided for TransNormerLLM models larger than 7B is provided, the performance on benchmarks is not included. It would be beneficial to include those results if possible.

**Questions:**

- Although the pre-training performance is on par with that of Transformer models, have any experiments on fine-tuning the models been conducted?
- In Algorithm 3, why is an explicit mask $M \in R^{N \times N}$ necessary? Is it for general attention?

---

> ### Author Response · Authors · 2023-11-20
> **Response to Reviewer LFwx**
>
> Q1. The meaning of exponential decay position encoding is mentioned in section 3.1.1.
>
> Exponential decay  position encoding indicates the term $\lambda^{s-t}$, where $\lambda$ $\in$(0,1). In this case, the QK attention scores are biased towards distanced tokens, with a proportional penalty for distance. In the first layer, it is combined with LRPE (Linearized Relative Positional Encoding) and is in the form of $a_{st}=\mathbf q_s^{\top} \mathbf k_t \lambda^{s-t}\exp^{i\theta(s-t)}$ where $\theta$ is the learnable parameter. While in the rest layers, it is in the form of $a_{st}=\mathbf q_s^{\top} \mathbf k_t \lambda^{s-t}$. In both cases, $\lambda$ is a non-learnable parameter to control the distance penalty in different layers as discussed in section 3.1.1 and equation(2). Such a form is fully compatible with Linear Attention, as it can be decomposed with respect to s and t separately. Thank you for the suggestion and we will clarify it in the revised version.
>
> Q2. Although the training efficiency for TransNormerLLM models larger than 7B is presented, the performance on benchmarks is not included. It would be advantageous to include those results if feasible.
>
> Thanks for your comments. Due to time constraints, we only include benchmark results for TransNormerLLM-385M, TransNormerLLM-1B, and TransNormerLLM-7B. As an initial exploration of scaling efficient LLM, we believe that 7B is a commonly used LLM size and is large enough to demonstrate that linear attention models can achieve better performance than transformers while being significantly faster. However, larger models, such as the 13B and 65B, are currently being trained. Once the training is completed, we will benchmark them. Thank you for your understanding and patience.
>
> Q3. Although the pre-training performance is on par with that of Transformer models, have any experiments on fine-tuning the models been conducted?
>
> The main objective of this paper is to scale up linear attention-based language models and confirm their effectiveness and efficiency. As most of our competitors use base models, we will only benchmark our base model to ensure fairness. However, we recognize that fine-tuning for specific applications is also crucial, and we plan to release our chat model soon.
>
> Q4. In Algorithm 3, why is an explicit mask $\mathbf M \in \mathbb R^{N \times N}$ necessary? Is it for general attention?
>
> Yes, it is necessary in causal attention. In Algorithm 3, the $\mathbf Q\mathbf K^{\top}$are calculated first as general attention, and the mask is used to achieve causal attention. However, in our efficient implementation, the explicit mask is no longer needed, and the model can achieve linear training and inference computation complexity. We provide an efficiency comparison of our method and Flash Attention 1/2 below, the units in the speed table are seconds, while the units in the memory table are in MB. Smaller values indicate better performance in both speed and memory:
>
> Speed table(seconds):
>
> Forward:
>
> | Seqlen | Lightning | Flash2 | Flash1 |
> | --- | --- | --- | --- |
> | 1024.0 | 0.255829 | 0.188746 | 0.282317 |
> | 2048.0 | 0.408914 | 0.459194 | 0.875625 |
> | 4096.0 | 0.808699 | 1.552662 | 3.094903 |
> | 8192.0 | 1.601662 | 5.735785 | 11.579776 |
> | 16384.0 | 3.197140 | 22.200319 | 44.999168 |
> | 32768.0 | 6.380953 | 86.720512 | 178.348038 |
>
> Backward:
>
> | Seqlen | Lightning | Flash2 | Flash1 |
> | --- | --- | --- | --- |
> | 1024.0 | 0.647602 | 0.526302 | 0.786245 |
> | 2048.0 | 1.268677 | 1.516766 | 2.427769 |
> | 4096.0 | 2.507520 | 4.948076 | 8.316556 |
> | 8192.0 | 4.990168 | 17.650484 | 30.684160 |
> | 16384.0 | 9.925859 | 66.620415 | 117.788673 |
> | 32768.0 | 19.840511 | 258.737152 | 461.466614 |
>
> Memory Table(MB):
>
> Forward:
>
> | Seqlen | Lightning | Flash2 | Flash1 |
> | --- | --- | --- | --- |
> | 1024.0 | 64.000488 | 80.500488 | 96.500977 |
> | 2048.0 | 128.000488 | 161.000488 | 193.000977 |
> | 4096.0 | 256.000488 | 322.000488 | 386.000977 |
> | 8192.0 | 512.000488 | 644.000488 | 772.000977 |
> | 16384.0 | 1024.000488 | 1288.000488 | 1544.000977 |
> | 32768.0 | 2048.000488 | 2576.000488 | 3088.000977 |
>
> Backward:
> | Seqlen | Lightning | Flash2 | Flash1 |
> | --- | --- | --- | --- |
> | 1024.0 | 166.400488 | 215.400488 | 215.400488 |
> | 2048.0 | 332.800488 | 430.800488 | 430.800488 |
> | 4096.0 | 665.600488 | 861.600488 | 861.600488 |
> | 8192.0 | 1331.200488 | 1723.200488 | 1723.200488 |
> | 16384.0 | 2662.400488 | 3446.400488 | 3446.400488 |
> | 32768.0 | 5324.800488 | 6892.800488 | 6892.800488 |

---

### Meta-Review · Area_Chair_NC66 · 2023-12-04

**Metareview:**

Reviewers find the paper does lot of tricks to scale training of Transnormer architecture proposed in earlier works. While some reviewers find the results strong and rate the paper positively, 2 of the reviewers rated paper negatively. Key concerns are 1) The paper is only combining existing techniques and has limited novelty. 2) All the results are on some un-specified dataset.

I also share these concerns. Given all the experiments are on un-specified dataset it is hard to evaluate the contributions of the paper. Further I find the paper only does limited study on all the new changes it introduced to Transnormer. I think doing a rigorous evaluation also on some public benchmarks for small scale models will help the paper. Hence I recommend rejection of the current draft.

**Justification For Why Not Higher Score:**

All experiments are on some un-specified dataset.

**Justification For Why Not Lower Score:**

Paper does good amount of work scaling Transnormer.

---

### Decision · Program_Chairs · 2024-01-16

Reject